# Health disparities, and health behaviours of older immigrants & native population in Norway

**Samera Azeem Qureshi** [1,2] *****, **Marte Kjøllesdal** [1,2], **Abdi Gele** [1,2]

**1** Unit for Migration & Health, Norwegian Institute of Public Health (NIPH), Oslo, Norway, **2** Institute of Public Health Science, Norwegian University of Life Sciences, Ås, Norway

* sameraazeem.qureshi@fhi.no

## Abstract

We aimed to investigate and compare activities of daily living (ADL), instrumental ADL (IADL), poor self-rated health and the health behaviours among immigrants and the native population in Norway. We present results from analysis of two Norwegian surveys, (Living Conditions Survey on Health from 2015, Living Conditions Survey among Immigrants 2016). Using logistic regression models, odds ratios were estimated for functional ability, self-reported health, and health behavior among immigrants, with Norwegian born being the reference category. The first model was controlled for age and gender and the second model was additionally adjusted for educational level. Our analysis included 5343 participants, 2853 men (913 immigrants), and 2481 women (603 immigrants), aged 45–79 years. The age-group 45–66 years includes n = 4187 (immigrants n = 1431, men n = 856; women n = 575) and 67–79 years n = 1147 (immigrants n = 85, men n = 57; women n = 28). The percentage of Norwegians having ≥ 14 years of education was 86%, as compared to 56% among immigrants. The percentage of immigrants with no education at all was 11%. The employment rate among the Norwegian eldest age group was nearly double (14%) as compared to the immigrant group. Adjusted for age, gender and education, immigrants had higher odds than Norwegian of ADL and IADL, chronic diseases and overweight. There were no differences between immigrants and Norwegians in prevalence of poor self-reported health and smoking. Overall elderly immigrants are worse-off than Norwegians in parameters of health and functioning. Knowledge about health and functioning of elderly immigrants can provide a basis for evidence-based policies and interventions to ensure the best possible health for a growing number of elderly immigrants. Furthermore, for a better surveillance, planning of programs, making policies, decisions and improved assessment and implementation, ADL and IADLs limitations should be included as a variable in public health studies.

## Introduction

Old age is often associated with functional decline and physical challenges that compromise the ability to perform necessary tasks required for daily living. Activities of daily living (ADL)

**Data Availability Statement:** a.) The data used in this study is not owned by the authors, therefore it cannot be shared publicly. Interested researchers may request access to the data from Norwegian Centre for Research data, Surveys: LKH_ 2015

("Living Conditions Survey on Health from 2015 ") and LKI_2016 ("Living Conditions Survey among Immigrants 2016") in Norway at nsd@nsd.no. b.) The authors made use of a secondary anonymized dataset collected by Statistics Norway and owned by Norwegian Centre for Research data. Because the dataset was anonymized, specific ethical approval was not required for this study and was waived by the Norwegian Centre for Research data. c.) A confidentiality agreement with the Norwegian Centre for Research Data was signed to use the data. d.) The authors conducted the analyses in accordance with the Norwegian Centre for Research Data's data protection regulations.

**Funding:** The author(s) received no specific funding for this work.

**Competing interests:** The authors have declared that no competing interests exist.

is an index which is used to measure functional capacity [1]. It is split into basic or personal ADL and instrumental ADL (IADL). ADL and IADL disabilities increase with age [2]. ADL encompasses the abilities necessary for basic functioning, such as ability to walk, dress, eat, toilet, shower [3], whereas IADL includes skills for community dwelling such as shopping, paying bills, answering the telephone, groceries, medicines [4]. ADLs and IADLs are essential for independent living and are predictors of morbidity and mortality in older populations.

The number of immigrants in Europe, born outside the EU-states increased from 2,48 million in 2010 to 3,22 million in 2018 [5]. There are around 95.7 million men and 68.1 million women migrant workers globally, these workers make up a large part of the labor force in Europe [6]. Due to several challenges such as language and culture, migrants usually perform hazardous low paid jobs which the natives do not take up [7–9]. These in turn lead to adverse health outcomes later in life which usually start at younger ages [6, 7, 10].

Increased migration over the last decades in Norway has created substantial ethnic diversity among old people in the country. Meeting the needs of this diverse population is a key challenge as it requires an adaptation of health and social policies to this new situation [11]. In Norway, the immigrant population is younger than the general population. Over half of immigrants are in the 20–44 year age group, compared to a third of the general population [12]. However, in a few years' time this population will be getting older thus requiring frequent and advanced medical care.

By 2060, Norway can expect over 1.3 million people over 70 years old and a doubling of the dependency ratio (the ratio of the dependent part of the workforce over the productive part) [13]. According to a report from the Norwegian Institute of health (NIPH), the current ratio of working-age population aged 16–64 years to people under 16 or over 64 years is 2:1 in Norway [14]. This will drop to 1.6 by 2060 and further to below 1.4 by the turn of the century [14].

Although the health of elderly migrants has been a global concern, research in Norway has been neglecting this group. The few studies that exist show that this group face linguistic and cultural barrier that prevent them from social integration. Poor health status in combination with poor proficiency in the Norwegian language in turn lead to challenges in communication with health personnel [15]. The gap in self-reported health between immigrants and the general population is largest among the elderly, with immigrants having considerably poorer health than others [16]. While 73% of non-immigrants aged 55–74 reported their health as good or very good, only 46% of immigrants with same age group categorized their health as good or very good [17].

Understanding the health status of elderly immigrants and the factors that affect their health is important to meet the health needs of this population in Norway. To our knowledge there has been no published study on the health status among older immigrants in Norway. We aimed to investigate and compare ADL, IADL, poor self-rated health and the health behaviours among immigrants and the native population in Norway, aged 45 years and above.

## Methods

We present results from analysis of two survey data, LKH_ 2015 ("Living Conditions Survey on Health from 2015 ") [18] and LKI_2016 ("Living Conditions Survey among Immigrants 2016") in Norway [16].

### LKH_2015

LKH_2015 is the latest survey on living conditions among the entire population, which has comparable health related questions as the living conditions survey among immigrants (LKI_2016). The living conditions survey on health was conducted with a representative

sample of persons aged 16 years and over, extracted from Statistics Norway's demographics population register. The sample was a stratified sample, with one stratum for each of the 19 counties of Norway. In each of the counties, 700 potential respondents were drawn, except in Oslo, the capital, where 1,400 were drawn. In total, the gross sample consisted of 14,000 potential respondents. This randomly selected sample were sent an invitation letter with a brief description of the intent of the project, and address to respond (email or telephone) in case they were willing to participate (consent), and a brochure explaining the project. Data collection was conducted between August & December 2015. Participants responded either by email or phone and indicated the channel they would prefer for the interview. The surveys covered several topics including demographic factors, family, housing, employment and the working environment, overall health, health disabilities, chronic diseases, social contact, discrimination, and Norwegian language proficiency. Many of the questions are based on the European Social Survey [19].

## LKI_2016

The participation criteria for LKI_2016 was being an immigrant (immigrants were defined as individuals born abroad with two foreign-born parents and four foreign-born grandparents) aged 16 years and above, living in Norway with a minimum of two years residence by 1st October 2015, and originally from Poland, Bosnia and Herzegovina, Kosovo, Turkey, Iraq, Iran, Afghanistan, Pakistan, Sri Lanka, Vietnam, Eritrea and Somalia. At the time of data collection, there were 214,000 immigrants from these countries living in Norway, making up almost one third of all immigrants [16]. From randomly selected immigrants, groups of 500–900 individuals from each country were sent an invitation letter with a brief description of the intent of the project, and address to respond (email or telephone) in case they were willing to participate (consent), and a brochure explaining the project [17]. Data collection was conducted between November 2015 and July 2016. In LKI_2016, 4435 men and women aged 16–74 years participated (response rate 54.4%) [16]. Non-responders were sent a reminder by e-mail and SMS after 4–8 weeks.

The data in both surveys was collected by computer-assisted telephone, or face-to-face interviews at a place and language of the participants' choice. More detailed description of sampling and data collection is available in the reports on the two surveys from Statistics Norway [16, 18].

**Ethics.** This dataset was collected by Statistics Norway and issued by the Norwegian Centre for Research Data. Ethical approval or consent from participants was not required for this study because the dataset was anonymized. We conducted the analyses in accordance with the Norwegian Centre for Research Data regulations.

**Variables.** *ADL & IADL*. In this article, we use two indicators to determine the extent of problems with functioning and disability; i) difficulty in at least one out of six activities of daily living (ADLs);ii) inability or difficulty in performing one of seven instrumental activities of daily living (IADLs) required for independent living. The ADL functions include walking across a room, getting in and out of bed, bathing or showering, eating (such as cutting up your food), dressing (including putting on shoes and socks) and using the toilet (including getting up or down). The IADL abilities include using a map to figure out how to get around in a strange place, preparing a hot meal, shopping for groceries, making telephone calls, taking medications, house or garden and managing money, such as paying bills and keeping track of expenses [20].

We generated two dichotomous variables "ADL" and "IADL" by using the "anymatch" function in Stata 16. For each we combined any of the responses corresponding to the variables as described above.

*Self-reported health.* Chronic health condition (including high blood pressure, diabetes, cancer, arthritis, stroke, lung disease, heart disease) was reported by the respondents, and operationalized as a dichotomous variable "no" or "yes (≥1 condition)". Self-reported health was categorized as very good, good, not so good, or fair, bad and very bad. We collapsed some of the categories together to generate a dichotomous variable; "good" (very good and good) and "fair/poor" (fair, poor and very poor).

*Health behaviour.* Usage of healthcare was determined by response to the question "How many times did you visit your doctor in the last 12 months?", dichotomized into "≥1 visit" and "no visits".

Respondents were asked to report their weight (without clothes and shoes) in kilograms and height (without shoes) in metres. From this, we calculated BMI ($kg/m^2$) and categorized it into "not overweight" (BMI <25) and "overweight" (BMI ≥ 25). Smoking is dichotomized as current smoker and non-smoker.

*Socioeconomic position.* We have categorized education into four categories, "No education", "Elementary school (≥10 years)", High school (11–13 years), University level (≥14 years). The categorization is in accordance with the educational system in Norway. Labor market participation is assessed by employment status as "employed" and "not employed".

Gender was categorized as men (0), women (1), while age-groups were categorized as "45–66 years" and "67–79 years". Countries of birth were "Norway", "Poland", "Turkey", "Bosnia-Herzegovina", "Kosovo", "Eritrea", "Somalia", "Afghanistan", "Sri-Lanka", "Iraq", "Iran", "Pakistan" or "Vietnam", with "Norway" as the reference category.

**Sample.** Literature on elder population normally includes age 50 years and above, but due to a small number of immigrants and structure of our data we had to lower the age limit in our sample to 45 years rather than 50 years.

## Statistical analyses

We ran logistic regression models to estimate odds ratios of functional ability, self-reported health and health behavior among immigrants, with Norwegian born being the reference category. The first model was controlled for age and gender to adjust for the compositional difference between the two groups independent of health. The second model was additionally adjusted for educational level to control for the socio-economic differences between the immigrants and Norwegians. We did not run regression analyses separately by country of origin due to very few numbers in individual categories. We used Stata v.16 for the analyses.

## Results

Our analysis included 5343 participants aged 45–79 years, 1516 (28.4%) were immigrants. The age-group 45–66 years included 4187 participants, 1431(37%) immigrants (men n = 856; women n = 575). Similarly, the age-group 67–79 years included 1147 participants with only 85 (7.4%) were immigrants (men n = 57; women n = 28). The percentage of Norwegians having ≥ 14 years of education was 86%, as compared to 56% among immigrants. The percentage of immigrants with no education at all was 11%. The employment rate among the Norwegian eldest age group was nearly double (14%) as compared to the immigrant group.

Table 1 presents the percent of the population reporting health problems among immigrants and non-immigrants and the variability across the different immigrant groups. There is a substantial variation in the prevalence of ADLs and IADLs between the immigrants and non-immigrants. Immigrants have almost five times higher prevalence of ADLs and three times higher prevalence of IADLs as non-immigrants. Norwegians rate their health high with only 24.1% reporting poor health. The percentage of immigrants reporting their health as bad

**Table 1. Percentage of participants with health problems, low self-reported health and risk factors.**

|  | Total | ADLs | IADLs | Chronic Diseases | Low self-reported Health | Overweight | Smokers |
|---|---|---|---|---|---|---|---|
| Immigrants | 1516 | 19.2 | 28.4 | 45.1 | 47.5 | 65.4 | 24.4 |
| **Country of Birth** |  |  |  |  |  |  |  |
| Norway | 3818 | 3.7 | 10.0 | 41.5 | 24.1 | 57.6 | 21.6 |
| Poland | 101 | 10.8 | 16.8 | 30.6 | 26.7 | 70.7 | 38.6 |
| Turkey | 123 | 15.4 | 29.2 | 59.3 | 55.2 | 87.7 | 37.4 |
| Bosnia | 171 | 22.8 | 31.5 | 33.9 | 42.6 | 72.0 | 30.9 |
| Kosovo | 120 | 27.5 | 40.8 | 48.3 | 50.8 | 76.5 | 35.8 |
| Eritrea | 76 | 17.1 | 15.7 | 32.9 | 32.8 | 44.6 | 11.8 |
| Somalia | 76 | 17.1 | 23.6 | 46.5 | 31.6 | 61.9 | 7.9 |
| Afghanistan | 36 | 30.5 | 33.3 | 55.5 | 52.7 | 70.9 | 22.2 |
| Sri Lanka | 216 | 17.1 | 19.4 | 40.3 | 49.1 | 54.0 | 10.1 |
| Iraq | 113 | 22.1 | 40.7 | 48.7 | 60.1 | 77.0 | 24.8 |
| Iran | 165 | 24.8 | 28.4 | 46.1 | 47.2 | 64.1 | 28.5 |
| Pakistan | 165 | 18.1 | 35.8 | 56.3 | 50.0 | 81.6 | 15.8 |
| Vietnam | 154 | 12.3 | 25.3 | 47.4 | 58.4 | 29.2 | 28.0 |

(47.5%) aligns with the presence of one or more chronic disease. A lower proportion of immigrants (74.2%) than Norwegians (83.3%) reported to have visited a doctor the previous year. Health behaviors also varied between the two groups with a higher proportion being overweight (65.4%) and reporting to be smokers (24.4%) among immigrants than non-immigrants. However, there is a huge variability across different immigrant groups in the prevalence of the health problems. The maximum variation is seen in being overweight (87.7–29.2 percent), followed by low self-reported health (60.1–26.7 percent) and smoking (37.4–7.9 percent).

Adjusted for age, gender and education, immigrants had higher odds than Norwegians of ADL and IADL, chronic diseases and overweight (Table 2). After adjustments, there were no differences between immigrants and Norwegians in prevalence of poor self-reported health and smoking.

## Discussion

In this study, we examined differences in functioning ability and health between immigrants and Norwegian aged 45 years and above. When health of elderly people deteriorates, it increases their care needs and their dependance on the health services, which places burden on

**Table 2. Odds ratios of functionality, health problems and risk factors among immigrants compared to Norwegians, controlled for age, gender and education.**

|  | ADLs | IADLs | Chronic Diseases | Low self-reported Health | Overweight | Smokers |
|---|---|---|---|---|---|---|
| **Norwegians** | 1 | 1 | 1 | 1 | 1 | 1 |
| **Immigrants** |  |  |  |  |  |  |
| **Model 1** | 6.88* | 4.27* | 1.42* | 0.32 | 1.28* | 1.00 |
| **Model 2** | 6.50* | 4.01* | 1.36* | 0.37 | 1.25* | 0.99 |

Model 1: Controlled for age and sex.

Model 2: Controlled for age, sex and education.

ADL: Difficulty performing at least one activity of daily living, IADL: Difficulty performing at least one instrumental activity of daily living

Chronic Diseases: One or more chronic diseases, Low Self-reported health: Less than good health, Smoking: Current smoker and Overweight: BMI> = 25.

* significant <0.01

the system. Thus, it is beneficial to maintain good health and high functioning even into old age, both for the elderly and for the society. In our study, immigrants overall appear to be worse-off both in functionality and self-rated health as compared to the natives. Moreover, they have higher proportions with chronic conditions and overweight.

The presence of functional disabilities at younger ages among immigrants also indicates stresses during the life course both pre & post-migration [10, 21] as well as experiences early in life which affect later health outcomes [7, 22]. Differences in immigration motivation and selection mechanisms might also be found between younger and older immigrants and within the older immigrant population [21, 23]. Cultural barriers especially communication and language barriers are an impediment in accessing healthcare services which add to deterioration of health over time [24, 25]. It is well proven that the health of immigrants deteriorates over time post-migration [26–28].

There is a well-established link between low socio-economic status, education and health status throughout the course of one's life [29, 30]. Low-socioeconomic status, risky health behaviors and health inequalities can increase the risk of disabilities in older age. This may be a result of poor health choices as well as lack of treatment opportunities [31]. A study reported association of fewer years of education and frequent money shortage with an increased prevalence of IADLs limitations [32]. ADL and IADL among immigrants may indicate more manual and strenuous labor during their lifetime than the native-born. This may be due to lack of choices in job selection [26], low education [32], and financial constraints [30]. Dangerous and excessive physical strain increases risks of injuries resulting in difficulties in daily and independent living. This fact has been substantiated by published literature, stating that the chronic conditions of immigrants from low-HDI (Human Development Index) countries tends to deteriorate faster than that of immigrants from richer countries.

This can be because immigrants from low-HDI countries tend to work in the so-called 3D jobs (dirty, dangerous and demanding/demeaning) once settled in the European countries [26, 33].

There is a clear difference in the socio-economic status between immigrants and non-immigrants in our survey data [34], with only 25% of immigrants working as managers and academicians (College, University), whereas the corresponding percentage in the general population is 54%. Immigrants are overrepresented in professions that do not require high educational levels such as cleaners, transport industry, auxiliary workers, construction industry etc [16, 34]. Almost 53% of immigrant men are working under risky conditions (noise, ergonomically unsafe, exposure to chemicals etc) as compared to 38% in the general population [16]. Similar findings have also been reported by studies from other parts of the world, that immigrants more often than others work in unfriendly and risky environments leading to health hazards [6, 9, 35]. However, there is a huge gender difference regarding the type of profession, as women are less involved in physical labor and more in jobs such as social care & health care workers [16, 34].

Level of difficulties in daily living can be subjective and may also reflect individual variation [4]. The physically demanding nature of the jobs results in fading of the Healthy migrant effect quicker, and accelerated deterioration of general health.

The increased number of immigrants with ADL and IADL can also be the result of cultural barriers such as language barriers, lack of knowledge of the new system, low education, and low health literacy. A number of studies have reported the association between low education and ADL and IADL [36]. In addition, studies have also shown that people with lower levels of education tend to have a more rapid health decline in old age [37]. However, educational level often shows, a weaker association with health in old age [38, 39], which might be the case in our finding of no effect of educational level. In our study SES did not have any effect on the

functional disabilities or chronic conditions. Similar findings of no effect of education on ADL has been reported by another study [40]. This indicates that the level of education does not determine the overall health in this sample. This has also been reported by a study analyzing data from 11 different European countries on the health status of older immigrants [20]. In addition, language barrier has been reported by many published studies an important indicator for difficulty in getting along in a society and thus exacerbate IADLs [41].

More immigrants have reported having one or more chronic disease as compared to the native population. This is in concordance with reports from several studies about the high prevalence of risk factors such as obesity, unhealthy diet, lack of physical inactivity [42–45]. In addition, immigrants self-reported poor health increases faster with age than among others, and that self-reported good health declines faster with number of health conditions than among Norwegian-born. Old age itself leads to increased occurrence of chronic diseases and comorbidities [46]. Immigrants already suffer from these diseases at a relatively younger age which increases the risk of both ADL and IADL. However, immigrants overall reported less contact with their doctors, despite having more chronic diseases. The less usage of health care especially among older immigrants has also been reported by other studies [47, 48].

Previous studies have reported that subjective perceptions such as poor self-rated health is an important risk factor for ADL and IADL [4, 49, 50]. More immigrants reported poor self-rated health as compared to natives. This may be the result of having more chronic diseases. However, there was no increased odds of having low self-reported health between the two groups, this may be a result of data limitation as in our data immigrants reporting low health versus those not reporting low health was almost same. This could also be a result of residual confounding however, we did control for the age, gender, and education.

Our study results have limitations. We had data about the immigrants at one point in time only. We do not know about those who return to their countries of origin after becoming ill, another selection process which may affect the observed results. In addition, those who are already suffering from disabilities both from the immigrants as well as the native population may chose not to participate thus underestimating the results.

Although Norway has relatively high levels of health indicators for the entire population this is not reflected in the results from our study among immigrants. The existence of significant health differentials between the native population and immigrants has been extensively addressed in literature [51]. The discussion on the requirements and demands of the immigrant population on the health and social services has been going on for some years now. The comparison of health of immigrants and the native population is somewhat difficult at the time of immigration due to the healthy migrant effect. However, health differences can be examined at later stages when health deteriorates increasing the requirement and dependance on the health services and places burden on the system.

In conclusion, the results from our study can be useful in expansion of the understanding of current health issues and at the same time improve baseline information on immigrant health. This can help policy-makers to predict the impact of growing immigration on the health and social security needs of a growing and aging immigrant population. Furthermore, for a better surveillance, planning of programs, making policies, decisions and improved assessment and implementation ADL and IADLs limitations should be included as a variable in public health studies. Consequently, we should implement social and health policies to better the negative health outcomes among older immigrant in Norway.

## Author Contributions

**Conceptualization:** Samera Azeem Qureshi.

Data curation: Samera Azeem Qureshi.

Formal analysis: Samera Azeem Qureshi.

Methodology: Samera Azeem Qureshi.

Software: Samera Azeem Qureshi.

Supervision: Samera Azeem Qureshi.

Writing – original draft: Samera Azeem Qureshi.

Writing – review & editing: Samera Azeem Qureshi, Marte Kjøllesdal, Abdi Gele.

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
