## [Decision Letter · Decision Letter 0]

23 Nov 2021

PONE-D-21-14206

Health disparities, and Health Behaviours of Older immigrants & Native population in Norway

PLOS ONE

Dear Dr. Qureshi,

Thank you for submitting your manuscript to PLOS ONE. After careful consideration, we feel that it has merit but does not fully meet PLOS ONE’s publication criteria as it currently stands. Therefore, we invite you to submit a revised version of the manuscript that addresses the points raised during the review process.

Apologies for the delay. It has been specially challenging to allocate suitable and available reviewers to assess this paper. Our reviewer highlights the value of your study, but asks for some essential revisions, including a broader discussion about methodological, practical and gender-related issues and implications of the study. Please consider addressing all them if you decide to revise the manuscript.

We look forward to receiving your revised manuscript.

Kind regards,

Sergio A. Useche, Ph.D.

Academic Editor

PLOS ONE

Journal Requirements:

3. Please change "female” or "male" to "woman” or "man" as appropriate, when used as a noun (see for instance https://apastyle.apa.org/style-grammar-guidelines/bias-free-language/gender).

5. We noticed you have some minor occurrence of overlapping text with the following previous publication(s), which needs to be addressed:

- https://www.mdpi.com/1660-4601/17/20/7387/html 

- http://docplayer.net/37706939-Health-of-immigrants-in-european-countries.html

- https://onlinelibrary.wiley.com/doi/10.1111/j.1747-7379.2008.00150.x 

The text that needs to be addressed involves the Discussion.

In your revision ensure you cite all your sources (including your own works), and quote or rephrase any duplicated text outside the methods section. Further consideration is dependent on these concerns being addressed.

Reviewers' comments:

Reviewer's Responses to Questions

**Comments to the Author**

1. Is the manuscript technically sound, and do the data support the conclusions?

Reviewer #1: Yes

2. Has the statistical analysis been performed appropriately and rigorously? 

Reviewer #1: Yes

3. Have the authors made all data underlying the findings in their manuscript fully available?

Reviewer #1: Yes

4. Is the manuscript presented in an intelligible fashion and written in standard English?

Reviewer #1: Yes

5. Review Comments to the Author

Reviewer #1: The study addresses a relevant and emerging issue/challenge in public health: health disparities between immigrants and native population in European countries, especially in old age that is associated with functional decline and physical challenges that, in turn, compromise the ability to perform necessary tasks required for daily living.

INTRODUCTION:

As the Authors stated in the introduction, there are some previous studies, although not many, dealing with the same topic.

I do suggest to the Authors to include in the introduction some comments about other previous references that, to my opinion, it is worth mentioning. For instance, IJERPH in 2019 published a special issue dealing with "Migration, Work and Health", which included both some original papers and a scoping review addressing the topic of immigrant workers' health disparities in European countries.

DISCUSSION:

I do suggest to the Authors to add some comments about gender differences (if any) and some more details (e.g. some examples taken from the included surveys) about the negative effect of "3-D" jobs on the health of immigrants.

6. PLOS authors have the option to publish the peer review history of their article (what does this mean?). If published, this will include your full peer review and any attached files.

Reviewer #1: No

---

## [Author Response · Author response to Decision Letter 0]

2 Jan 2022

Response Reviewers

Journal Requirements:

Response: Done.

Response: Done.

3. Please change "female” or "male" to "woman” or "man" as appropriate, when used as a noun (see for instance https://apastyle.apa.org/style-grammar-guidelines/bias-free-language/gender).

Response: Done

Response: As we have indicated in our previously published article from the same survey data “Qureshi SA, Straiton M, Gele AA. Associations of socio-demographic factors with adiposity among immigrants in Norway: a secondary data analysis. BMC Public Health. 2020 May 24;20(1):772. doi: 10.1186/s12889-020-08918-9. PMID: 32448125; PMCID: PMC7247236.

Availability of data and materials: The data is available from NSD (Norwegian Centre for Research

Data) upon request nsd@nsd.no.

Ethics approval and consent to participate: The data is available from NSD (Norwegian Centre for Research Data) upon request. We signed a privacy contract before we started analysing the data. The survey was approved by the regional ethical committee and the participants gave written consent. The

participants were free to withdraw at any time and their data was deleted immediately with no consequences.

Consent for publication: As our manuscript does not include any individual data or sensitive personal information, therefore consent for publication is “Not Applicable” in this case.

5. We noticed you have some minor occurrence of overlapping text with the following previous publication(s), which needs to be addressed:

- https://www.mdpi.com/1660-4601/17/20/7387/html

- http://docplayer.net/37706939-Health-of-immigrants-in-european-countries.html

- https://onlinelibrary.wiley.com/doi/10.1111/j.1747-7379.2008.00150.x

The text that needs to be addressed involves the Discussion.

Response: We have rectified the overlapping text.

Abstract: Line 52-54.

Discussion: pg 9, line235-236. Pg 13, line 318-322. Pg 13, line 323-333.

In your revision ensure you cite all your sources (including your own works), and quote or rephrase any duplicated text outside the methods section. Further consideration is dependent on these concerns being addressed.

Reviewers' comments:

Reviewer's Responses to Questions

Comments to the Author

1. Is the manuscript technically sound, and do the data support the conclusions?

Reviewer #1: Yes

2. Has the statistical analysis been performed appropriately and rigorously? 

Reviewer #1: Yes

3. Have the authors made all data underlying the findings in their manuscript fully available?

Reviewer #1: Yes

4. Is the manuscript presented in an intelligible fashion and written in standard English?

Reviewer #1: Yes

5. Review Comments to the Author

Reviewer #1: The study addresses a relevant and emerging issue/challenge in public health: health disparities between immigrants and native population in European countries, especially in old age that is associated with functional decline and physical challenges that, in turn, compromise the ability to perform necessary tasks required for daily living.

INTRODUCTION:

As the Authors stated in the introduction, there are some previous studies, although not many, dealing with the same topic.

I do suggest to the Authors to include in the introduction some comments about other previous references that, to my opinion, it is worth mentioning. For instance, IJERPH in 2019 published a special issue dealing with "Migration, Work and Health", which included both some original papers and a scoping review addressing the topic of immigrant workers' health disparities in European countries.

Response: We appreciate the response of the reviewer and have now added additional comments in the introduction pg.3 line 68-73 “The number of immigrants in Europe, born outside the EU-states increased from 2,48 million in 2010 to 3,22 million in 2018 (5). There are around 95.7 million men and 68.1 million women migrant workers globally, these workers are a large part of the labor force particularly in Europe (6). Due to several challenges such as language and culture, migrants usually perform hazardous low paid jobs which the natives do not take up (7-9). These in turn lead to adverse health outcomes later in life which usually start at younger ages (6, 7, 10)”. 

And on pg 3 line 83-85 “According to a report from the Norwegian Institute of health (NIPH), today the ratio is 2:1 i.e two people working (16-64 years) for every person under 16 or over 64 years (14). This will drop to 1.6 by 2060 and further to below 1.4 by the turn of the century (14)”.

DISCUSSION:

I do suggest to the Authors to add some comments about gender differences (if any) and some more details (e.g. some examples taken from the included surveys) about the negative effect of "3-D" jobs on the health of immigrants.

Response: Again we agree with the reviewers comments and have now added additional text in the discussion section on pg. 11, lines 264-274

“There is a clear difference in the socio-economic status in our survey data (34), according to which only 25% of immigrants work as managers, academicians (College, University), whereas the corresponding percentage in the general population is 54%. Immigrants are overrepresented in professions that do not require high educational levels such as cleaners, transport industry, auxiliary workers, construction industry etc (16, 34). Almost 53% of immigrant men have to work under risky conditions (noise, ergonomically unsafe, exposure to chemicals etc) as compared to 38% in the general population (16). This has also been reported by studies from other parts of the world, that immigrants have to work in unfriendly and risky environments leading to health hazards (6, 9, 35). However, there is a huge gender difference regarding the type of profession, as women are less involved in physical labor and more in jobs such as social care & health care workers (16, 34)”. 

6. PLOS authors have the option to publish the peer review history of their article (what does this mean?). If published, this will include your full peer review and any attached files.

Do you want your identity to be public for this peer review? For information about this choice, including consent withdrawal, please see our Privacy Policy.

Reviewer #1: No

---

## [Decision Letter · Decision Letter 1]

18 Jan 2022

Health disparities, and Health Behaviours of Older immigrants & Native population in Norway

PONE-D-21-14206R1

Dear Dr. Qureshi,

We’re pleased to inform you that your manuscript has been judged scientifically suitable for publication and will be formally accepted for publication once it meets all outstanding technical requirements.

Kind regards,

Sergio A. Useche, Ph.D.

Academic Editor

PLOS ONE

Additional Editor Comments (optional):

Reviewers' comments:

Reviewer's Responses to Questions

**Comments to the Author**

1. If the authors have adequately addressed your comments raised in a previous round of review and you feel that this manuscript is now acceptable for publication, you may indicate that here to bypass the “Comments to the Author” section, enter your conflict of interest statement in the “Confidential to Editor” section, and submit your "Accept" recommendation.

Reviewer #1: All comments have been addressed

2. Is the manuscript technically sound, and do the data support the conclusions?

Reviewer #1: (No Response)

3. Has the statistical analysis been performed appropriately and rigorously? 

Reviewer #1: (No Response)

4. Have the authors made all data underlying the findings in their manuscript fully available?

Reviewer #1: (No Response)

5. Is the manuscript presented in an intelligible fashion and written in standard English?

Reviewer #1: (No Response)

6. Review Comments to the Author

Reviewer #1: (No Response)

7. PLOS authors have the option to publish the peer review history of their article (what does this mean?). If published, this will include your full peer review and any attached files.

Reviewer #1: No

---

## [Editor Report · Acceptance letter]

21 Jan 2022

PONE-D-21-14206R1 

Health disparities, and Health Behaviours of Older immigrants & Native population in Norway 

Dear Dr. Qureshi:

I'm pleased to inform you that your manuscript has been deemed suitable for publication in PLOS ONE. Congratulations! Your manuscript is now with our production department. 

Kind regards, 

on behalf of

Dr. Sergio A. Useche 

Academic Editor

PLOS ONE